# Neural speech restoration at the cocktail party: Auditory cortex recovers masked speech of both attended and ignored speakers

**Christian Brodbeck**[1] *, **Alex Jiao**[2], **L. Elliot Hong**[3], **Jonathan Z. Simon**[1,2,4]

**1** Institute for Systems Research, University of Maryland, College Park, Maryland, United States of America, **2** Department of Electrical and Computer Engineering, University of Maryland, College Park, Maryland, United States of America, **3** Maryland Psychiatric Research Center, Department of Psychiatry, University of Maryland School of Medicine, Baltimore, Maryland, United States of America, **4** Department of Biology, University of Maryland, College Park, Maryland, United States of America

* christianbrodbeck@me.com

**Data Availability Statement:** Preprocessed MEG recordings and stimuli are available from the Digital Repository at the University of Maryland. The MEG dataset is available at http://hdl.handle.net/1903/

## Abstract

Humans are remarkably skilled at listening to one speaker out of an acoustic mixture of several speech sources. Two speakers are easily segregated, even without binaural cues, but the neural mechanisms underlying this ability are not well understood. One possibility is that early cortical processing performs a spectrotemporal decomposition of the acoustic mixture, allowing the attended speech to be reconstructed via optimally weighted recombinations that discount spectrotemporal regions where sources heavily overlap. Using human magnetoencephalography (MEG) responses to a 2-talker mixture, we show evidence for an alternative possibility, in which early, active segregation occurs even for strongly spectrotemporally overlapping regions. Early (approximately 70-millisecond) responses to nonoverlapping spectrotemporal features are seen for both talkers. When competing talkers' spectrotemporal features mask each other, the individual representations persist, but they occur with an approximately 20-millisecond delay. This suggests that the auditory cortex recovers acoustic features that are masked in the mixture, even if they occurred in the ignored speech. The existence of such noise-robust cortical representations, of features present in attended as well as ignored speech, suggests an active cortical stream segregation process, which could explain a range of behavioral effects of ignored background speech.

## Introduction

When listening to an acoustic scene, the signal that arrives at the ears is an additive mixture of the different sound sources. Listeners trying to selectively attend to one of the sources face the task of determining which spectrotemporal features belong to that source [1]. When multiple speech sources are involved, as in the classic cocktail party problem [2], this is a nontrivial problem because the spectrograms of the different sources often have strong overlap. Nevertheless, human listeners are remarkably skilled at focusing on one out of multiple talkers [3,4].

21109, additional files specific to this paper are available at http://hdl.handle.net/1903/26370. Subject-specific results are provided for each figure in supplementary data files.

**Funding:** This work was supported by a National Institutes of Health grant R01-DC014085 (to JZS; https://www.nih.gov) and by a University of Maryland Seed Grant (to LEH and JZS; https://umd.edu). The funders had no role in study design, data collection and analysis, decision to publish, or preparation of the manuscript.

**Competing interests:** The authors have declared that no competing interests exist.

**Abbreviations:** EEG, electroencephalography; HG, Heschl's gyrus; MEG, magnetoencephalography; ROI, region of interest; SNR, signal-to-noise ratio; STG, superior temporal gyrus; STRF, spectrotemporal response function; TRF, temporal response function.

Binaural cues can facilitate segregation of different sound sources based on their location [5] but are not necessary for this ability, because listeners are able to selectively attend even when 2 speech signals are mixed into a monophonic signal and presented with headphones [6]. Here we are specifically interested in the fundamental ability to segregate and attend to one out of multiple speakers even without such external cues.

The neural mechanisms involved in this ability are not well understood, but previous research suggests at least 2 separable cortical processing stages. In magnetoencephalography (MEG) responses to multiple talkers [7], the early (approximately 50-millisecond) cortical component is better described as a response to the acoustic mixture than as the sum of the responses to the individual (segregated) source signals, consistent with an early unsegregated representation of the mixture. In contrast, the later (> 85 millisecond) response component is dominated by the attended (segregated) source signal. Recent direct cortical recordings largely confirm this picture, suggesting that early responses in Heschl's gyrus (HG) reflect a spectro-temporal decomposition of the acoustic mixture that is largely unaffected by selective atten-tion, whereas later responses in the superior temporal gyrus (STG) dynamically change to represent the attended speaker [8]. In general, cortical regions further away from core auditory cortex tend to mainly reflect information about the attended speaker [9]. Together, these results suggest a cortical mechanism that, based on a detailed representation of the acoustic input, detects and groups features belonging to the attended source.

A long-standing question is whether early cortical processing of the acoustic mixture is restricted to passive spectrotemporal filtering, or whether it involves active grouping of acous-tic features leading to the formation of auditory object representations. The filter theory of attention suggests that early representations reflect physical stimulus characteristics indepen-dent of attention, with attention selecting a subset of these for further processing and semantic identification [10,11]. Consistent with this, electroencephalography (EEG) and MEG results suggest that time-locked processing of higher order linguistic features, such as words and meaning, is restricted to the attended speech source [12,13]. However, it is not known whether, in the course of recovering the attended source, the auditory cortex also extracts acoustic fea-tures of the ignored source from the mixture. Individual intracranially recorded HG responses to a 2-speaker mixture can be selective for either one of the speakers, but this selectivity can be explained merely by spectral response characteristics favoring the spectrum of a given speaker over the other [8]. A conservative hypothesis is thus that early auditory cortical responses rep-resent acoustic features of the mixture based on stable (possibly predefined) spectrotemporal receptive fields, allowing the attended speech to be segregated through an optimally weighted combination of these responses. Alternatively, the auditory cortex could employ more active mechanisms to dynamically recover potential speech features, regardless of what stream they belong to. Selective attention could then rely on these local auditory (proto-) objects to recover the attended speech [14]. This hypothesis critically predicts the existence of representations of acoustic features from an ignored speech source, even when those features are not apparent in the acoustic mixture, i.e., when those features are masked by acoustic energy from another source. Here we report evidence for such representations in human MEG responses.

Cortical responses to speech reflect a strong representation of the envelope (or spectro-gram) of the speech signal [15,16]. Prior work has also shown that acoustic onsets are promi-nently represented in auditory cortex, both in naturalistic speech [17,18] and in nonspeech stimuli [19,20]. Studies using paradigms similar to the one used here often predicted brain responses from only envelopes or only onsets [16,21–24], but more recent studies show that both representations explain nonredundant portions of the responses [12,18]. Behaviorally, acoustic onsets are also specifically important for speech intelligibility [25,26]. Here we con-sider both envelope and onset features but focus on onset features in particular because of

their relevance for stream segregation, as follows [1]. If acoustic elements in different frequency regions are co-modulated over time, they likely stem from the same physical source [27]. A simultaneous onset in distinct frequency bands thus provides sensory evidence that these cross-frequency features originate from the same acoustic source and should be processed as an auditory object. Accordingly, shared acoustic onsets promote perceptual grouping of acoustic elements into a single auditory object, such as a complex tone and, vice versa, separate onsets lead to perceptual segregation [28,29]. For example, the onset of a vowel is characterized by a shared onset at the fundamental frequency of the voice and its harmonics. Correspondingly, if the onset of a formant is artificially offset by as little as 80 milliseconds, it can be perceived as a separate tone rather than as a component of the vowel [30]. This link to object perception thus makes acoustic onsets particularly relevant cues, which might be represented distinctly from envelope cues and used to detect the beginning of local auditory objects, and thus aid segregation of the acoustic input into different, potentially overlapping auditory objects.

We analyzed human MEG responses to a continuous 2-talker mixture to determine to what extent the auditory cortex reliably tracks acoustic onset or envelope features of the ignored speech, above and beyond the attended speech and the mixture. Participants listened to 1-minute-long continuous audiobook segments, spoken by a male or a female speaker. Segments were presented in 2 conditions: a single talker in quiet ("clean speech"), and a 2-talker mixture, in which a female and a male speaker were mixed at equal perceptual loudness. MEG responses were analyzed as additive, linear response to multiple concurrent stimulus features (see Fig 1). First, cross-validated model comparisons were used to determine which representations

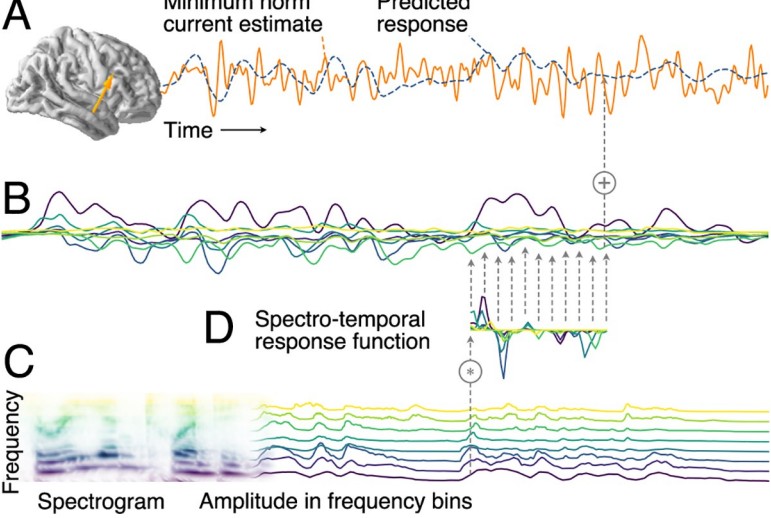

**Fig 1. Additive linear response model based on STRFs.** (A) MEG responses recorded during stimulus presentation were source localized with distributed minimum norm current estimates. A single virtual source dipole is shown for illustration, with its physiologically measured response and the response prediction of a model. Model quality was assessed by the correlation between the measured and the predicted response. (B) The model's predicted response is the sum of tonotopically separate response contributions generated by convolving the stimulus envelope at each frequency (C) with the estimated TRF of the corresponding frequency (D). TRFs quantify the influence of a predictor variable on the response at different time lags. The stimulus envelopes at different frequencies can be considered a collection of parallel predictor variables, as shown here by the gammatone spectrogram (8 spectral bins); the corresponding TRFs as a group constitute the STRF. Physiologically, the component responses (B) can be thought of as corresponding to responses in neural subpopulations with different frequency tuning, with MEG recording the sum of those currents. MEG, magnetoencephalographic; STRF, spectrotemporal response function; TRF, temporal response function.

significantly improve prediction of the MEG responses. Then, the resultant spectrotemporal response functions (STRFs) were analyzed to gain insight into the nature of the representations.

## Results and discussion

### Auditory cortex represents acoustic onsets

MEG responses to clean speech were predicted from the gammatone spectrogram of the stimulus and, simultaneously, from the spectrogram of acoustic onsets (Fig 2A). Acoustic onsets were derived from a neural model of auditory edge detection [19]. The 2 predictors were each binned into 8 frequency bands, such that the MEG responses were predicted from a model of the acoustic stimulus encompassing 16 time series in total. Each of the 2 predictors was assessed based on how well (left-out) MEG responses were predicted by the full model, compared with a null model in which the relevant predictor was omitted. Both predictors significantly improve predictions (onsets: $t_{max} = 12.00$, $p \leq 0.001$; envelopes: $t_{max} = 9.39$, $p \leq 0.001$), with an anatomical distribution consistent with sources in HG and STG bilaterally (Fig 2B). Because this localization agrees with findings from intracranial recordings [8,17], results were henceforth analyzed in an auditory region of interest (ROI) restricted to these 2 anatomical landmarks (Fig 2C). When averaging the model fits in this ROI, almost all subjects showed evidence of responses associated with both predictors (Fig 2D).

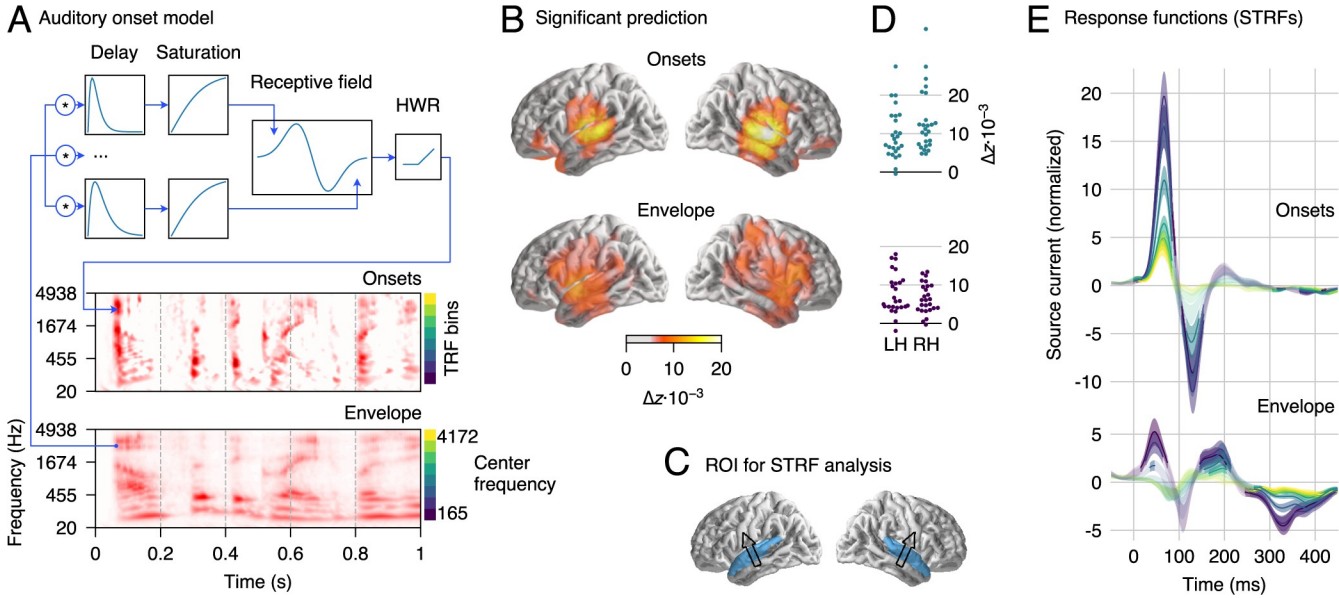

**Fig 2. MEG responses to clean speech.** (A) Schematic illustration of the neurally inspired acoustic edge detector model, which was used to generate onset representations. The signal at each frequency band was passed through multiple parallel pathways with increasing delays, so that an "edge detector" receptive field could detect changes over time. HWR removed the negative sections to yield onsets only. An excerpt from a gammatone spectrogram ("envelope") and the corresponding onset representation are shown for illustration. (B) Regions of significant explanatory power of onset and envelope representations, determined by comparing the cross-validated model fit from the combined model (envelopes + onsets) to that when omitting the relevant predictor. Results are consistent with sources in bilateral auditory cortex ($p \leq 0.05$, corrected for whole brain analysis). (C) ROI used for the analysis of response functions, including superior temporal gyrus and Heschl's gyrus. An arrow indicates the average dominant current direction in the ROI (upward current), determined through the first principal component of response power. (D) Individual subject data corresponding to (B), averaged over the ROI in the LH and RH, respectively. (E) STRFs corresponding to onset and envelope representations in the ROI; the onset STRF exhibits a clear pair of positive and negative peaks, while peaks in the envelope STRF are less well-defined. Different color curves reflect the frequency bins, as indicated next to the onset and envelope spectrograms in panel A. Shaded areas indicate the within-subject standard error (SE) [31]. Regions in which STRFs differ significantly from 0 are marked with more saturated (less faded) colors ($p \leq 0.05$, corrected for time/frequency). Data are available in S1 Data. HWR, half-wave rectification; LH, left hemisphere; MEG, magnetoencephalographic; RH, right hemisphere; ROI, region of interest; SE, standard error; STRF, spectrotemporal response function; TRF, temporal response function.

Auditory cortical STRFs were summarized for each subject and hemisphere using a subject-specific spatial filter based on principal component analyses of overall STRF power in the ROI. The average direction of that spatial filter replicates the direction of the well-known auditory MEG response (Fig 2C, arrows). This current vector is consistent with activity in core auditory cortex and the superior temporal plane. However, MEG sensors are less sensitive to radial currents, as would be expected from lateral STG areas implicated by intracranial recordings [8]. Because of this, we focus here on the temporal information in STRFs rather than drawing conclusions from the spatial distribution of sources. STRFs can thus be interpreted as reflecting different processing stages associated with different latencies, possibly involving multiple regions in the superior temporal lobe. STRFs were initially separately analyzed by hemisphere, but because none of the reported results interact significantly with hemisphere, the results shown are collapsed across hemisphere to simplify presentation.

STRFs to acoustic onsets exhibit a well-defined 2-peaked shape, consistent across frequency bands (Fig 2E). An early, positive peak (average latency 65 milliseconds) is followed by a later, negative peak (126 milliseconds). This structure closely resembles previously described auditory response functions to envelope representations when estimated without consideration of onsets [16]. In comparison, envelope STRFs in the present results are diminished and exhibit a less well-defined structure. This is consistent with acoustic onsets explaining a large portion of the signal usually attributed to the envelope; indeed, when the model was refitted with only the envelope predictor, excluding the onset predictor, the envelope STRFs exhibited that canonical pattern and with larger amplitudes (see S1 Fig).

STRFs have disproportionately higher amplitudes at lower frequencies (Fig 2E), which is consistent with previous tonotopic mapping of speech areas and may follow from the spectral distribution of information in the speech signal [32,33]. This explanation is also supported by simulations, where responses to speech were generated using equal temporal response functions (TRFs) for each band, and yet estimated STRFs exhibited higher amplitudes in lower frequency bands (see S1 Simulations, Fig S1).

## Auditory cortex represents ignored speech

MEG responses to a 2-speaker mixture were then analyzed for neural representations of ignored speech. Participants listened to a perceptually equal loudness mixture of a male and a female talker and were instructed to attend to one talker and ignore the other. The speaker to be attended was counterbalanced across trials and subjects. Responses were predicted using both onset and envelope representations for: the acoustic mixture, the attended speech source, and the ignored source (Fig 3A). The underlying rationale is that, because the brain does not have direct access to the individual speech sources, if there is neural activity corresponding to either source separately (above and beyond the mixture), this indicates that cortical responses have segregated or reconstructed features of that source from the mixture. Both predictors representing the ignored speech significantly improve predictions of the responses in the ROI (both $p < 0.001$, onsets: $t_{max} = 6.70$, envelopes: $t_{max} = 6.28$; group level statistics were evaluated with spatial permutation tests; subject-specific model fits, averaged in the ROI are shown in Fig 3B). This result indicates that acoustic features of the ignored speech are represented neurally even after controlling for features of the mixture and the attended source. The remaining 4 predictors also significantly increased model fits (all $p < 0.001$; mixture onsets: $t_{max} = 8.61$, envelopes: $t_{max} = 5.70$; attended onsets: $t_{max} = 6.32$, envelopes: $t_{max} = 7.37$).

Onset STRFs exhibit the same characteristic positive–negative pattern as for responses to a single talker but with reliable distinctions between the mixture and the individual speech streams (Fig 3C and 3D). The early, positive peak occurs earlier and has a larger amplitude for

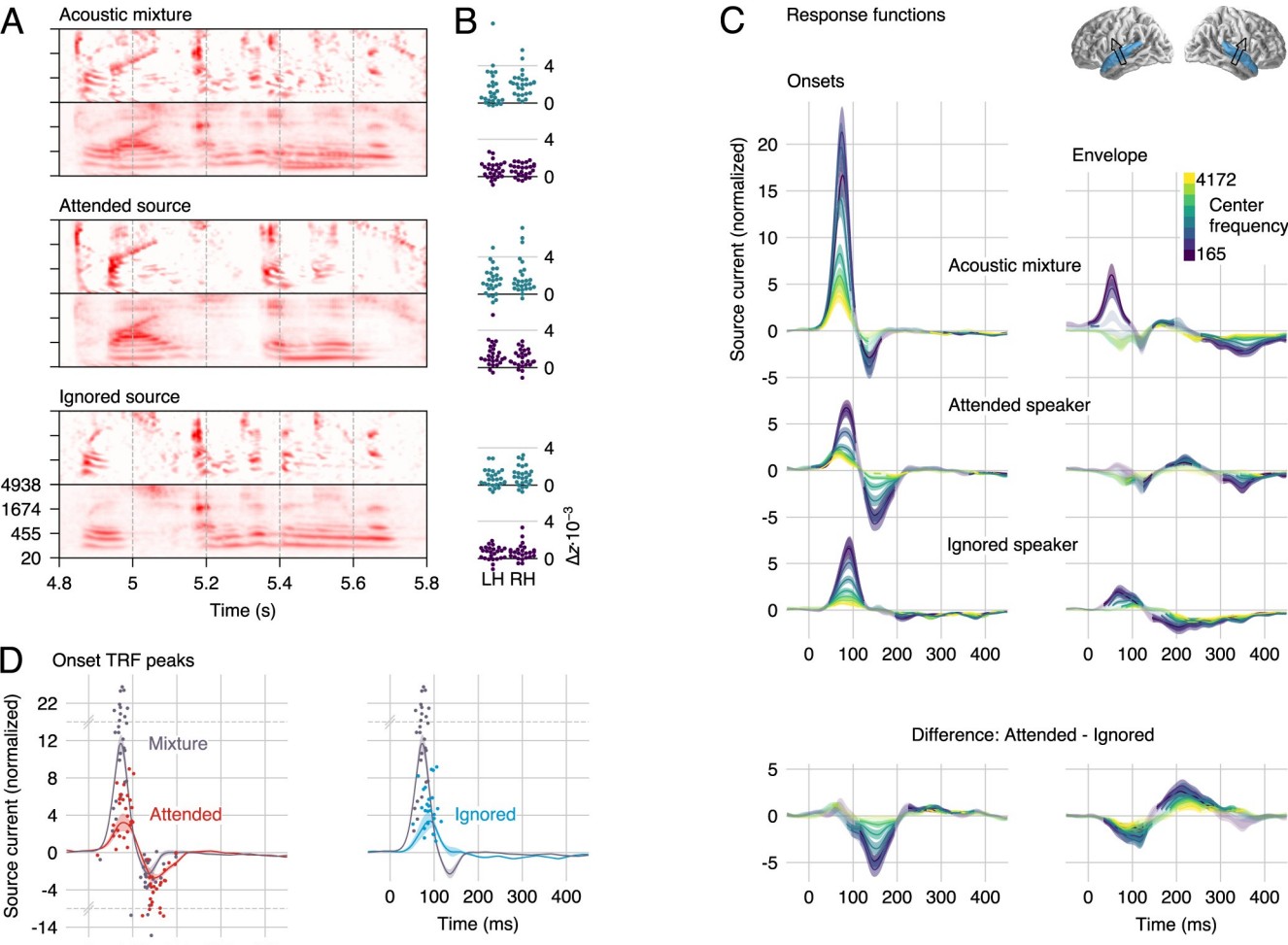

**Fig 3. Responses to the 2-speaker mixture, using the stream-based model.** (A) The envelope and onset representations of the acoustic mixture and the 2 speech sources were used to predict MEG responses. (B) Individual subject model fit improvement due to each predictor, averaged in the auditory cortex ROI. Each predictor explains neural data not accounted for by the others. (C) Auditory cortex STRFs to onsets are characterized by the same positive/negative peak structure as STRFs to a single speaker. The early, positive peak is dominated by the mixture but also contains speaker-specific information. The second, negative peak is dominated by representations of the attended speaker and, to a lesser extent, the mixture. As with responses to a single talker, the envelope STRFs have lower amplitudes, but they do show a strong and well-defined effect of attention. Explicit differences between the attended and ignored representations are shown in the bottom row. Details as in Fig 2. (D) The major onset STRF peaks representing individual speech sources are delayed compared with corresponding peaks representing the mixture. To determine latencies, mixture-based and individual-speaker-based STRFs were averaged across frequency (lines with shading for mean ±1 SE). Dots represent the largest positive and negative peak for each subject between 20 and 200 milliseconds. Note that the y-axis is scaled by an extra factor of 4 beyond the indicated break points at y = 14 and −6. Data are available in S2 Data. LH, left hemisphere; MEG, magnetoencephalography; RH, right hemisphere; ROI, region of interest; SE, standard error; STRF, spectrotemporal response function.

onsets in the mixture than for onsets in either of the sources (latency mixture: 72 milliseconds; attended: 81 milliseconds, $t_{25} = 4.47$, $p < 0.001$; ignored: 89 milliseconds, $t_{25} = 6.92$, $p < 0.001$; amplitude mixture > attended: $t_{25} = 8.41$, $p < 0.001$; mixture > ignored: $t_{25} = 7.66$, $p < 0.001$). This positive peak is followed by a negative peak only in responses to the mixture (136 milliseconds) and the attended source (150 milliseconds; latency difference $t_{25} = 3.20$, $p = 0.004$). The amplitude of these negative peaks is statistically indistinguishable ($t_{25} = 1.56$, $p = 0.132$).

The mixture predictor is not completely orthogonal to the source predictors. This might raise a concern that a true response to the mixture might cause spurious responses to the sources. Simulations using the same predictors as used in the experiment suggest, however, that such contamination is unlikely to have occurred (see S1 Simulations).

## Envelope processing is strongly modulated by selective attention

Although the envelope STRFs seem to be generally less structured than those of the onsets, a comparison of the STRFs to the attended and the ignored source revealed a strong and well-defined effect of attention (Fig 3C, right column). The attended-ignored difference wave exhibits a negative peak at approximately 100 milliseconds, consistent with previous work [16], and an additional positive peak at approximately 200 milliseconds. In contrast with previous work, however, a robust effect of attention on the envelope representation starts almost as early as the very earliest responses. Thus, accounting for responses to onset features separately reveals that envelope processing is thoroughly influenced by attention. The reason for this might be that onsets often precede informative regions in the spectrogram, such as the spectral detail of voiced segments. The onsets might thus serve as cues to direct attention to specific regions in the spectrogram [28], which would allow early attentional processing of the envelope features.

## Auditory cortex "un-masks" masked onsets

The analysis using the stream-based predictors suggests that the auditory cortex represents acoustic onsets in both speech sources separately, in addition to onsets in the acoustic mixture. This is particularly interesting because, while envelopes combine mostly in an additive manner, acoustic onsets may experience negative interference. This can be seen in the spectrograms in Fig 3A: The envelope mixture representation largely looks like a sum of the 2 stream envelope representations. In contrast, the onset mixture representation has several features that have a lower amplitude than the corresponding feature in the relevant source. The finding of a separate neural representation of the source onsets thus would suggest that the auditory cortex reconstructs source features that are masked in the mixture. Such reconstruction might be related to instances of cortical filling-in, in which cortical representations show evidence of filling in missing information to repair degraded, or even entirely absent, input signals [34–36]. The latency difference between mixture and source onsets might then reflect a small additional processing cost for the recovery of underlying features that are not directly accessible in the sensory input.

However, the specifics of what we call mixture and source features depends to some degree on the model of acoustic representations, i.e., the gammatone and edge detection models used here. Specifically, source features that are here masked in the mixture might be considered overt in a different acoustic model. It is unlikely that all our source features are, in reality, overt, because then our mixture representation should not be able to predict any brain responses beyond the acoustic sources. However, the apparent neural representations of stream-specific onsets could be of a secondary set of features that the mixture is transparent to. An example could be a secondary stage of onset extraction based on pitch; the delay in responses to source specific onsets might then simply reflect the latency difference of spectral and pitch-based onset detection.

Although these 2 possibilities could both explain the results described so far, they make different predictions regarding responses to masked onsets. A passive mechanism, based on features to which the mixture is transparent, should be unaffected by whether the features are masked in the gammatone representation, because the masking does not actually apply to those features. Such responses should thus be exhaustively explained by the stream-based model described in Fig 3. On the other hand, an active mechanism that specifically processes masked onsets might generate an additional response modulation for masked onsets. To test for such a modulation, we subdivided the stream-based onset predictors to allow for different responses to overt and masked onsets. The new predictors were implemented as element-wise operations on the onset spectrograms (Fig 4A). Specifically, for each speech source, the new

"masked onsets" predictor models the degree to which an onset in the source is attenuated (masked) in the mixture, i.e., the amount by which a segregated speech source exceeds the (physically presented) resultant acoustic mixture, or zero when it does not: (*max(source–mixture, 0)*). The new "overt onsets" predictor models all other time-frequency points, where an onset in the source is also seen as a comparable onset in the mixture (element-wise *min(mixture, source)*). Note that with this definition the sum of overt and masked onsets exactly equals the original speech source onset representation, i.e., the new predictors model the same onsets but allow responses to differ depending on whether an onset is masked or not. Replacing the 2 speech-source-based onset predictors with the 4 overt/masked onset predictors significantly improves the model fit ($t_{max} = 6.81$, $p < 0.001$), suggesting that cortical responses indeed distinguish between overt and masked onsets. Each of the 4 new predictors individually contributes to the MEG responses, although masked onsets do so more robustly (attended: $t_{max} = 8.42$, $p < 0.001$; ignored: $t_{max} = 5.23$, $p < .001$) than overt onsets (attended: $t_{max} = 3.34$, $p = 0.027$; ignored: $t_{max} = 3.82$, $p = 0.016$; Fig 4B); this difference could be due to overt source onsets being more similar to the mixture onsets predictor. Critically, the significant effect for masked onsets in the ignored source confirms that the auditory cortex recovers masked onsets even when they occur in the ignored source.

## Masked onsets are processed with a delay and an early effect of attention

Model comparison thus indicates that the neural representation of masked onsets is significantly different from that of overt onsets. The analysis of STRFs suggests that this is for at least 2 reasons (Fig 4C–4E): response latency differences and a difference in the effect of selective attention.

First, responses to masked onsets are systematically delayed compared with overt onsets (as can be seen in Fig 4D). Specifically, this is true for the early, positive peak (mixture: 72 milliseconds), both for the attended speaker (overt: 72 milliseconds, masked: 91 millisecond, $t_{25} = 2.85$, $p = 0.009$) and the ignored speaker (overt: 83 milliseconds, masked: 97 millisecond, $t_{25} = 6.11$, $p < 0.001$). It is also the case for the later, negative peak (mixture: 138 milliseconds), which reflects only the attended speaker (overt: 133 milliseconds, masked: 182 milliseconds, $t_{25} = 4.45$, $p < 0.001$). Thus, at each major peak, representations of masked onsets lag behind representations of overt onsets by at least 15 milliseconds.

Second, for overt onsets, the early representations at the positive peak appear to be independent of the target of selective attention (Fig 4C, top right). In contrast, for masked onsets, even these early representations are enhanced by attention (Fig 4C, bottom right). This difference is confirmed in a stream (attended, ignored) by masking (overt, masked) ANOVA on peak amplitudes with a significant interaction ($F_{(1,25)} = 24.45$, $p < 0.001$). For overt onsets, Fig 4E might suggest that the early peak is actually enhanced for the ignored speaker; however, this difference can be explained by the early onset of the second, negative response to attended onsets, which overlaps with the earlier peak. This observation makes the early effect of attention for masked onsets all the more impressive, because the early peak is larger despite the onset of the subsequent, negative peak (note the steeper slope between positive and negative peak for attended masked onsets). Also note that we here interpreted the timing of the effect of attention relative to the peak structure of the TRFs; in terms of absolute latency, the onset of the effect of attention is actually more similar between masked and overt onsets (see Fig 4E).

## Delayed response to masked onsets

Previous research has found that the latency of responses to speech increases with increasing levels of stationary noise [37,38] or dynamic background speech (Fig 3 in [21]). Our results

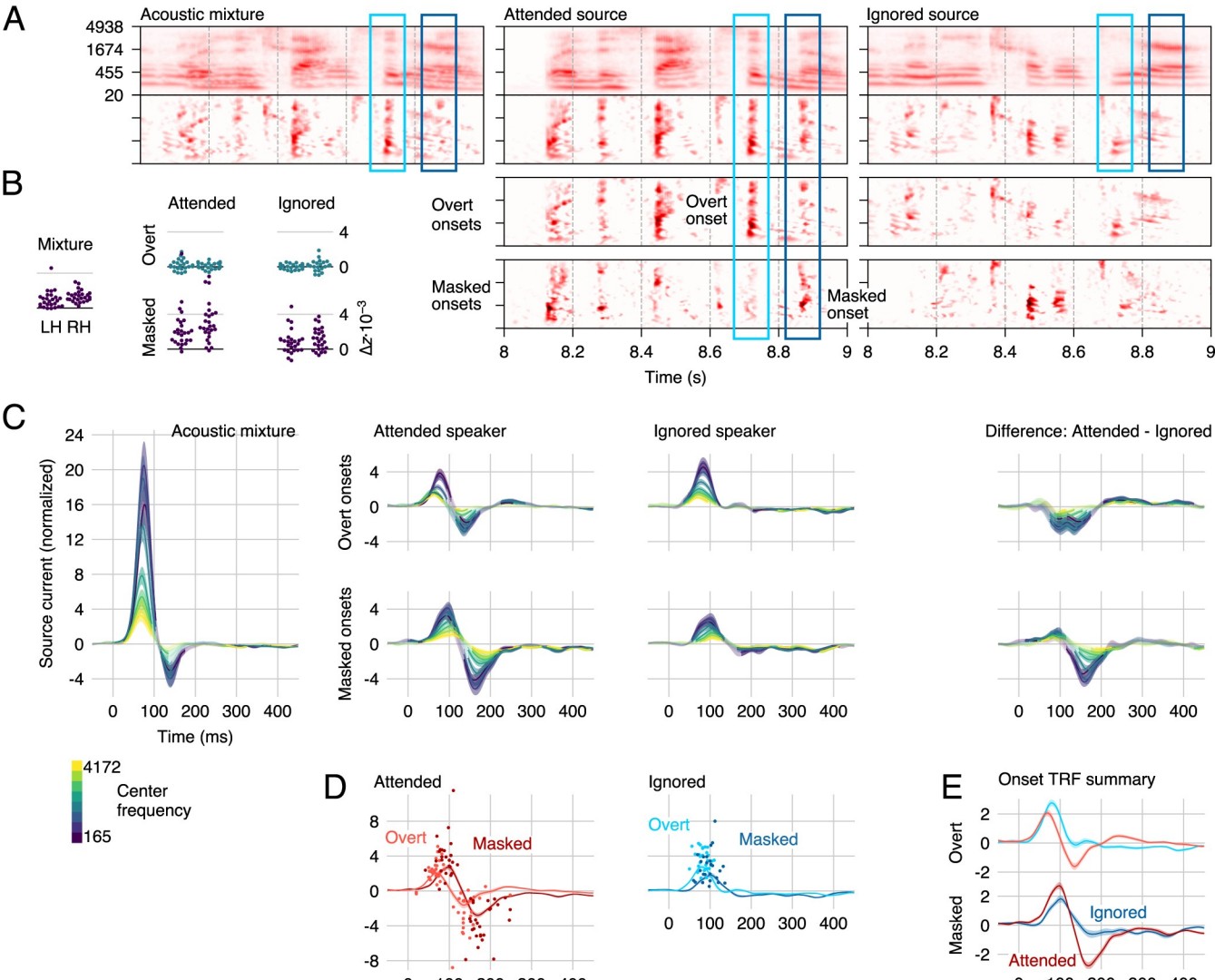

**Fig 4. Responses to overt and masked onsets.** (A) Spectrograms (note that in this Fig, the onset representations are placed below the envelope representations, to aid visual comparison of the different onset representations) were transformed using element-wise operations to distinguish between overt onsets, i.e., onsets in a source that are apparent in the mixture, and masked onsets, i.e., onsets in a source that are masked in the presence of the other source. Two examples are marked by rectangles: The light blue rectangle marks a region with an overt (attended) onset, i.e., an onset in the attended source that also corresponds to an onset in the mixture. The dark blue rectangle marks a masked (attended) onset, i.e., an onset in the attended source which is not apparent in the mixture. (B) All predictors significantly improve the cross-validated model fit (note that improvements were statistically tested with a test sensitive to spatial variation, whereas these plots show single-subject ROI average fits). (C) The corresponding overt/masked STRFs exhibit the previously described positive–negative 2-peaked structure. The first, positive peak is dominated by a representation of the mixture but also contains segregated features of the 2 talkers. For overt onsets, only the second, negative peak is modulated by attention. For masked onsets, even the first peak exhibits a small degree of attentional modulation. (D) Responses to masked onsets are consistently delayed compared with responses to overt onsets. Details are analogous to Fig 3D, except that the time window for finding peaks was extended to 20–250 milliseconds to account for the longer latency of masked onset response functions. (E) Direct comparison of the frequency-averaged onset TRFs highlights the amplitude differences between the peaks. For overt onsets, the negative deflection due to selective attention starts decreasing the response magnitude even near the maximum of the first, positive peak. For masked onsets, the early peak reflecting attended onsets is increased despite the subsequent enhanced negative peak. Results for envelope predictors are omitted from this figure because they are practically indistinguishable from those in Fig 3. Data are available in S3 Data. LH, left hemisphere; RH, right hemisphere; ROI, region of interest; STRF, spectrotemporal response function; TRF, temporal response function.

indicate that, for continuous speech, this is not simply a uniform delay but that the delay varies dynamically for each acoustic element based on whether this element is overt or locally masked by the acoustic background. This implies that the stream is not processed as a homogeneous

entity of constant signal-to-noise ratio (SNR) but that the acoustic elements related to the onsets constitute separate auditory objects, with processing time increasing dynamically as features are more obscured acoustically. Notably, the same applies to the ignored speech stream, suggesting that acoustic elements from both speakers are initially processed as auditory objects.

The effect of SNR on response amplitude and latency is well established and clearly related to the results here. We consider it unlikely that SNR can itself play the role of a causal mechanistic explanation, because the measure of SNR presupposes a signal and noise that have already been segregated. Consequently, SNR is not a property of features in the acoustic input signal. Acoustic features only come to have an SNR after they are designated as acoustic objects and segregated against an acoustic background, such that their intensity can be compared with that of the residual background signal. This is illustrated in our paradigm in that the same acoustic onset can be a signal for one process (when detecting onsets in the mixture) and part of the noise for another (when attending to the other speaker). Rather than invoking SNR itself as an explanatory feature, we thus interpret the delay as evidence for a feature detection mechanism that requires additional processing time when the feature in question is degraded in the input—although leaving open the specific mechanism by which this happens. The addition of stationary background noise to simple acoustic features is associated with increased response latencies to those features as early as the brainstem response wave-V [39]. This observed shift in latency is in the submillisecond range and may have a mechanistic explanation in terms of different populations of auditory nerve fibers: Background noise saturates high spontaneous rate fibers, and the response is now dominated by somewhat slower, low spontaneous rate fibers [40]. In cortical responses to simple stimuli, like tones, much larger delays are observed in the presence of static noise, in the order of tens of milliseconds [41].

Latency shifts due to absolute signal intensity [42] might be additive with shifts due to noise [43]. Such a nonlinear increase in response latency with intensity might be a confounding factor in our analysis, which is based on linear methods: Compared with overt onsets, masked onsets in the ignored talker should generally correspond to weaker onsets in the mixture. Splitting overt and masked onsets might thus improve the model fit because it allows modeling different response latencies for different intensity levels of onsets in the mixture, rather than reflecting a true response to the background speech. In order to control for this possibility, we compared the model fit of the background-aware model with a model allowing for 3 different intensity levels of onsets in the mixture (and without an explicit representation of onsets in the ignored speaker). The background speaker-aware model outperformed the level-aware model ($t_{max}$ = 9.21, $p$ < 0.001), suggesting that the present results are not explained by this level-based nonlinear response to the mixture. Furthermore, a background-unaware nonlinearity would not explain the difference in the effect of attention between overt and masked onsets. Together, this suggests that the observed delay is related to recovering acoustic source information, rather than a level-based nonlinearity in the response to the mixture.

It is also worth noting that the pattern observed in our results diverges from the most commonly described pattern of SNR effects. Typically, background noise causes an amplitude decrease along with the latency increase [38]: Although the latency shift observed here conforms to the general pattern, the amplitude of responses to masked onsets is not generally reduced. Even more importantly, selective attention affects the delayed responses more than the undelayed, suggesting that the delay is not a simple effect of variable SNR but is instead linked to attentive processing. A study of single units in primary auditory cortex found that neurons with delayed, noise-robust responses exhibited response properties suggestive of network effects [44]. This is consistent with the interaction of selective attention and delay found

here on the early peak, because the delayed responses to masked onsets also exhibit more evidence of goal-driven processing than the corresponding responses to overt onsets.

In sum, masking causes latency increases at different stages of the auditory system. These latency shifts increase at successive stages of the ascending auditory pathway, as does the preponderance of noise-robust response properties [45]. It is likely that different levels of the auditory system employ different strategies to recover auditory signals of interest and do so at different time scales. Together with these considerations, our results suggest that the auditory cortex actively recovers masked speech features, and not only of the attended, but also of the ignored speech source.

## Early effect of selective attention

Besides the shift in latency, response functions to overt and masked onsets differed in a more fundamental way: While the early, positive response peak to overt onsets did not differentiate between attended and ignored onsets, the early peak to masked onsets contained significantly larger representations of attended onsets (see Fig 4C). Thus, not only do early auditory cortical responses represent masked onsets, but these representations are substantively affected by whether the onset belongs to the attended or the ignored source. This distinction could have several causes. In the extreme, it could indicate that the 2 streams are completely segregated and represented as 2 distinct auditory objects. However, it might also be due to a weighting of features based on their likelihood of belonging to the attended source. This could be achieved, for example, through modulation of excitability based on spectrotemporal prediction of the attended speech signal [46]. Thus, onsets that are more likely to belong to the attended source might be represented more strongly, without yet being ascribed to one of the sources exclusively.

One discrepancy in previous studies using extra- and intracranial recordings is that the former were unable to detect any early effects of selective attention [7,16], whereas the latter showed a small but consistent enhancement of feature representations associated with the attended acoustic source signal [8]. Furthermore, an early effect of attention would also be expected based on animal models that show task-dependent modulations of A1 responses [47,48]. Our results show that this discrepancy may depend, in part, on which acoustic features are analyzed: While overt acoustic onsets were not associated with an effect of selective attention, masked onsets were.

Overall, the early difference between the attended and ignored source suggests that acoustic information from the ignored source is represented to a lesser degree than information from the attended source. This is consistent with evidence from psychophysics suggesting that auditory representations of background speech are not as fully elaborated as those of the attended foreground [49]. More generally, it is consistent with results that suggest an influence of attention early on in auditory stream formation [50].

## Stages of speech segregation through selective attention

Regardless of whether listening to a single talker or 2 concurrent talkers, response functions to acoustic onsets are characterized by a prominent positive–negative 2 peak structure. Brain responses to 2 concurrent talkers allow separating these response functions into components related to different representations and thus reveal different processing stages. Fig 5 presents a plausible model incorporating these new findings. Early on, the response is dominated by a representation of the acoustic mixture, with a preliminary segregation, possibly implemented through spectral filters [8]. This is followed by restoration of speech features that are masked in the mixture, regardless of speaker, but with a small effect of selective attention, suggesting a

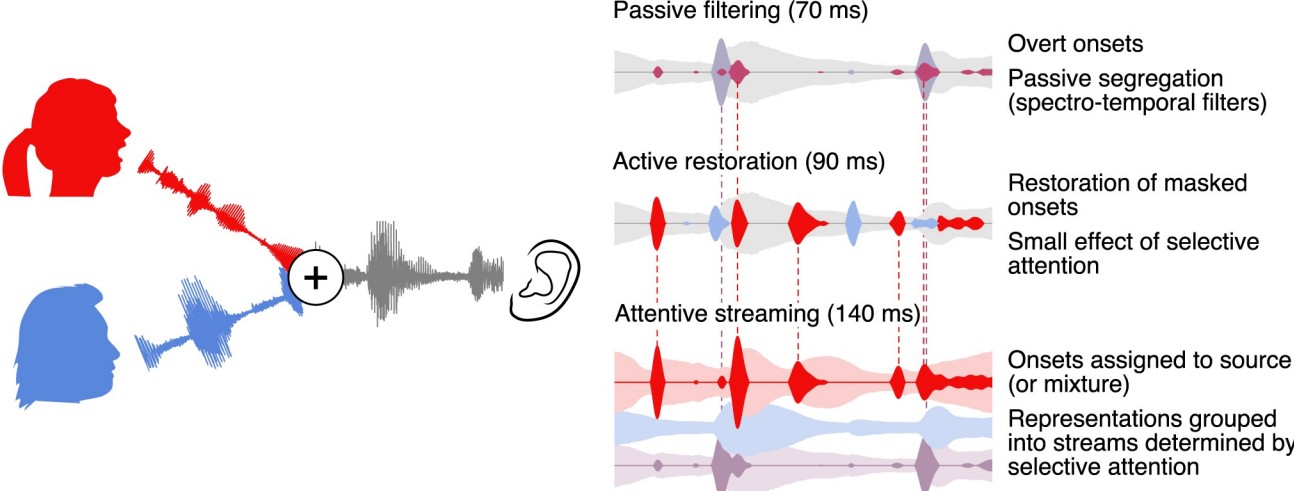

**Fig 5. Model of onset-based stream segregation.** A model of cortical processing stages compatible with the results reported here. Left: The auditory scene, with additive mixture of the waveforms from the attended and the ignored speakers (red and blue, respectively). Right: Illustration of cortical representations at different processing stages. Passive filtering: At an early stage, onsets are extracted from the acoustic mixture and representations are partially segregated, possibly based on frequency. This stage corresponds to the early positive peak in onset TRFs. Active Restoration: A subsequent stage also includes representations of onsets in the underlying speech sources that are masked in the mixture, corresponding to the first peak in TRFs to masked onsets. At this stage, a small effect of attention suggests a preliminary selection of onsets with a larger likelihood of belonging to the attended speaker. Streaming: Finally, at a third stage, the response to onsets from the ignored speaker is suppressed, suggesting that now the 2 sources are clearly segregated (see also [8]). This stage corresponds to the second, negative peak, which is present in TRFs to mixture and attended onsets but not to ignored onsets. TRF, temporal response function.

more active mechanism. Finally, later responses are dominated by selective attention, suggesting a clearly separated representation of the attended speaker as would be expected in successful streaming.

An open question concerns how the overt and masked feature representations are temporally integrated. Representations of masked onsets were consistently delayed compared with those of overt onsets by approximately 20 milliseconds (see Fig 4D). This latency difference entails that upstream speech processing mechanisms may receive different packages of information about the attended speech source with some temporal desynchronization. Although this might imply a need for a higher order corrective mechanism, it is also possible that upstream mechanisms are tolerant to this small temporal distortion. A misalignment of 20 milliseconds is small compared with the normal temporal variability encountered in speech (although phonetic contrasts do exist where a distortion of a few tens of milliseconds would be relevant). Indeed, in audio-visual speech perception, temporal misalignment up to 100 milliseconds between auditory and visual input can be tolerated [51].

Broadly, the new results are consistent with previous findings that early cortical responses are dominated by the acoustic mixture, rather than receiving presegregated representations of the individual streams [7,8]. However, the new results do show evidence of an earlier, partial segregation, in the form of representations of acoustic onsets, which are segregated from the mixture, though not grouped into separate streams. Because these early representations do not strictly distinguish between the attended and the ignored speaker, they likely play the role of an intermediate step in extracting the information needed to selectively attend to one of the 2 speakers. Overall, these results are highly consistent with object-based models of auditory attention, in which perception depends on an interplay between bottom-up analysis and formation of local structure, and top-down selection and global grouping, or streaming [14,52].

## Implications for processing of "ignored" acoustic sources

The interference in speech perception from a second talker can be very different from the interference caused by nonspeech sounds. For instance, music is cortically segregated from speech even when both signals are unattended, consistent with a more automatic segregation, possibly due to distinctive differences in acoustic signal properties [22]. In contrast, at moderate SNRs, a second talker causes much more interference with speech perception than a comparable nonspeech masker. Interestingly, this interference manifests not just in the inability to hear attended words but in intrusions of words from the ignored talker [53]. The latter fact, in particular, has been interpreted as evidence that ignored speech might be segregated and processed to a relatively high level. On the other hand, listeners seem to be unable to process words in more than 1 speech source at a time, even when the sources are spatially separated [54]. Furthermore, demonstrations of lexical processing of ignored speech are rare and usually associated with specific perceptual conditions such as dichotic presentation [55]. Consistent with this, recent EEG/MEG evidence suggests that unattended speech is not processed in a time-locked fashion at the lexical [12] or semantic [13] level. The results described here, showing systematic recovery of acoustic features from the ignored speech source, suggest a potential explanation for the increased interference from speech compared with other maskers. Representing onsets in 2 speech sources could be expected to increase cognitive load compared with detecting onsets of a single source in stationary noise. These representations of ignored speech might also act as bottom-up cues and cause the tendency for intrusions from the ignored talker. They might even explain why a salient and overlearned word, such as one's own name [56], might sometimes capture attention, which could happen based on acoustic rather than lexical analysis [57]. Finally, at very low SNRs, the behavioral pattern can invert, and a background talker can be associated with better performance than stationary noise maskers [53]. In such conditions, there might be a benefit of being able to segregate the ignored speech source and use this information strategically [21].

An open question is how the auditory system deals with the presence of multiple background speakers. When there are multiple background speakers, does the auditory system attempt to unmask the different speakers all separately, or are they represented as a unified background [7]? An attempt to isolate speech features even from multiple background talkers might contribute to the overly detrimental effect of babble noise with a small number of talkers [58].

## Limitations

Many of the conclusions drawn here rest on the suitability of the auditory model used to predict neural responses. The gammatone and onset models are designed to reflect generalized cochlear and neural processing strategies and were chosen as more physiologically realistic models than engineering-inspired alternatives such as envelope and half-wave rectified derivative models. Yet they might also be missing critical aspects of truly physiological representations. An important consideration for future research is thus to extend the class of models of lower level auditory processing and how they relate to the large-scale neural population activity as measured by EEG/MEG.

In addition, our model incorporates the masking of acoustic features as a binary distinction, by splitting features into overt and masked features. In reality, features can be masked to degrees. In our model, intermediate degrees of maskedness would result in intermediate values in both predictors and thus, in a linear superposition of 2 responses. We would expect that a model that could take into account the degree of maskedness as continuous variable would likely provide a better fit to the neural data.

## Conclusions

How do listeners succeed in selectively listening to one of 2 concurrent talkers? Our results suggest that active recovery of acoustic onsets plays a critical role. Early responses in the auditory cortex represent not only overt acoustic onsets but also reconstruct acoustic onsets in the speech sources that are masked in the mixture, even if they originate from the ignored speech source. This suggests that early responses, in addition to representing a spectrotemporal decomposition of the mixture, actively reconstruct acoustic features that could originate from either speech source. Consequently, these early responses make comparatively complex acoustic features from both speech sources available for downstream processes, thus enabling both selective attention and bottom-up effects of salience and interference.

## Materials and methods

### Participants

The data analyzed here have been previously used in an unrelated analysis [12] and can be retrieved from the Digital Repository at the University of Maryland (see Data Availability). MEG responses were recorded from 28 native speakers of English, recruited by media advertisements from the Baltimore area. Participants with medical, psychiatric, or neurological illnesses, head injury, and substance dependence or abuse were excluded. Data from 2 participants were excluded, one due to corrupted localizer measurements and one due to excessive magnetic artifacts associated with dental work, resulting in a final sample of 18 male and 8 female participants with mean age 45.2 (range 22–61).

### Ethics statement

All participants provided written informed consent in accordance with the University of Maryland Baltimore Institutional Review Board and were paid for their participation.

### Stimuli

Two chapters were selected from an audiobook recording of A Child's History of England by Charles Dickens, one chapter read by a male and one by a female speaker (https://librivox.org/a-childs-history-of-england-by-charles-dickens/, chapters 3 and 8, respectively). Four 1-minute long segments were extracted from each chapter (referred to as male-1 through 4 and female 1 through 4). Pauses longer than 300 milliseconds were shortened to an interval randomly chosen between 250 and 300 milliseconds, and loudness was matched perceptually (such that either speaker was deemed equally easy to attend to). Two-talker stimuli were generated by additively combining 2 segments, one from each speaker, with an initial 1-second period containing only the to-be attended speaker (mix-1 through 4 were constructed by mixing male-1 and female-1, through 4).

### Procedure

During MEG data acquisition, participants lay supine and were instructed to keep their eyes closed to minimize ocular artifacts and head movement. Stimuli were delivered through foam pad earphones inserted into the ear canal at a comfortably loud listening level, approximately 70 dB SPL.

Participants listened 4 times to mix-1 while attending to one speaker and ignoring the other (which speaker they attended to was counterbalanced across participants), then 4 times to mix-2 while attending to the other speaker. After each segment, participants answered a question relating to the content of the attended stimulus. Then, the 4 segments just heard were all

presented once each, as single talkers. The same procedure was repeated for stimulus segments 3 and 4.

## Data acquisition and preprocessing

Brain responses were recorded with a 157 axial gradiometer whole head MEG system (KIT, Kanazawa, Japan) inside a magnetically shielded room (Vacuumschmelze GmbH & Co. KG, Hanau, Germany) at the University of Maryland, College Park. Sensors (15.5-mm diameter) are uniformly distributed inside a liquid-He dewar, spaced approximately 25 mm apart, and configured as first-order axial gradiometers with 50 mm separation and sensitivity better than 5 fT·Hz$^{-1/2}$ in the white-noise region ($> 1$ KHz). Data were recorded with an online 200-Hz low-pass filter and a 60-Hz notch filter at a sampling rate of 1 kHz.

Recordings were preprocessed using mne-python (https://github.com/mne-tools/mne-python) [59]. Flat channels were automatically detected and excluded. Extraneous artifacts were removed with temporal signal space separation [60]. Data were filtered between 1 and 40 Hz with a zero-phase FIR filter (mne-python 0.15 default settings). Extended infomax independent component analysis [61] was then used to remove ocular and cardiac artifacts. Responses time-locked to the onset of the speech stimuli were extracted and resampled to 100 Hz. For responses to the 2-talker mixture, the first second of data, in which only the to-be attended talker was heard, was discarded.

Five marker coils attached to participants' head served to localize the head position with respect to the MEG sensors. Two measurements, one at the beginning and one at the end of the recording, were averaged. The FreeSurfer (https://surfer.nmr.mgh.harvard.edu) [62] "fsaverage" template brain was coregistered to each participant's digitized head shape (Polhemus 3SPACE FASTRAK) using rotation, translation, and uniform scaling. A source space was generated using 4-fold icosahedral subdivision of the white matter surface, with source dipoles oriented perpendicularly to the cortical surface. Minimum $\ell 2$ norm current estimates [63,64] were computed for all data. Initial analysis was performed on the whole brain as identified by the FreeSurfer "cortex" label. Subsequent analyses were restricted to sources in the STG and Heschl's gyrus as identified in the "aparc" parcellation [65].

## Predictor variables

Predictor variables were based on gammatone spectrograms sampled at 256 frequencies, ranging from 20 to 5,000 Hz in ERB space [66], resampled to 1 kHz and scaled with exponent 0.6 [67].

Acoustic onset representations were computed by applying an auditory edge detection model [19] independently to each frequency band of the spectrogram. The model was implemented with a delay layer with 10 delays ranging from $\tau_2 = 3$ to 5 milliseconds, a saturation scaling factor of $C = 30$, and a receptive field based on the derivative of a Gaussian window with $SD = 2$ input delay units. Negative values in the resulting onset spectrogram were set to 0. We initially explored using higher levels of saturation (smaller values for C) but found that the resulting stimulus representations emphasized nonspeech features during pauses more than features relevant to speech processing, because responses quickly saturated during ongoing speech. We chose the given, narrow range of $\tau_2$ to allow for a wider possibility of models because a wider range of $\tau_2$ would only lead to smoother representations, although smoothing can also achieved by the TRF model fitted to the neural data.

Onset representations of the 2 speakers were split into masked and overt onsets using element-wise operations on the onset spectrograms. Masked onsets were defined by the extent to

which onsets were larger in the source than in the mixture:

$$o_{masked} = \max(o_{source} - o_{mixture}, 0)$$

Overt onsets were onsets that were not masked, i.e., speech source onsets that were also visible in the mixture:

$$o_{overt} = o_{source} - o_{masked} \equiv min(o_{source}, o_{mixture})$$

Using this procedure, approximately 67% of total onset magnitudes ($\ell$1 norm) was assigned to overt onsets and 33% to masked onsets.

For model estimation, envelope and onset spectrograms were then binned into 8 frequency bands equally spaced in ERB space (omitting frequencies below 100 Hz because the female speaker had little power below that frequency) and resampled to match the MEG data. As part of the reverse correlation procedure, each predictor time series (i.e., each frequency bin) was scaled by its $\ell$1 norm over time.

For testing an intensity-based nonlinear response (see "Delayed response to masked onsets"), the onset predictor was split into 3 separate predictors, one for each of 3 intensity levels. For each of the 8 frequency bins, individual onsets were identified as contiguous nonzero elements; Each onset was assigned an intensity based on the sum of its elements, and the onsets were then assigned to one of 3 predictors based on intensity tertiles (calculated separately for each band). This resulted in three 8-band onset spectrograms modeling low-, medium-, and high-intensity onsets.

## Reverse correlation

STRFs were computed independently for each virtual current source [see 68]. The neural response at time $t$, $y_t$, was predicted from the sum of $N$ predictor variables $x_n$ convolved with a corresponding response function $h_n$ of length $T$:

$$\hat{y}_t = \sum_{n}^{N} \sum_{\tau}^{T} h_{n,\tau} \cdot x_{n,t-\tau}$$

STRFs were generated from a basis of 50-millisecond-wide Hamming windows and were estimated using an iterative coordinate descent algorithm [69] to minimize the $\ell$1 error.

For model evaluation, left-out data were predicted using 4-fold cross-validation. Folds were created by assigning successive trials to the different folds in order (1, 2, 3, 4, 1, 2, . . .). In an outer loop, the responses in each fold were predicted with STRFs estimated from the remaining 3 folds. These predictions, combined, served to calculate the correlation between measured and predicted responses used for model tests. In an inner loop, each of the 3 estimation folds was, in turn, used as validation set for STRFs trained on the 2 remaining folds. STRFs were iteratively improved based on the maximum error reduction in the training set (the steepest coordinate descent) and validated in the validation set. Whenever a predictor time series (i.e., one spectrogram bin) would have caused an increasing in the error in the validation set, the kernel for this predictor was frozen, continuing until all predictors were frozen (see [70] for further details). The 3 STRFs from the inner loop were averaged to predict responses in the left-out testing data.

## Model tests

Each spectrogram comprising 8 time series (frequency bins) was treated as an individual predictor. Speech in quiet was modeled using the (envelope) spectrogram and acoustic onsets:

$$MEG \sim o + e$$

where $o$ = onsets and $e$ = envelope. Models were estimated with STRFs with $T = [0,...,500)$ millisecond. Model quality was quantified through the Pearson correlation $r$ between actual and predicted responses. The peak of the averaged $r$-map was 0.143 in the single-speaker condition and 0.158 in the 2-talker condition (0.162 when the model incorporated masking). Because single-trial MEG responses contain a relatively high proportion of spontaneous and nonneural (and hence unexplainable) signals, the analysis focused on differences between models that are reliable across participants, rather than absolute $r$-values. Each model was thus associated with a map of Fisher $z$-scored $r$-values, smoothed with a Gaussian kernel (SD = 5 mm). In order to test the predictive power of each predictor, a corresponding null model was generated by removing that predictor. For each predictor, the model quality of the full model was compared with the model quality of the corresponding null model using a mass-univariate related measures $t$-test with threshold-free cluster enhancement [71] and a null distribution based on 10,000 permutations. This procedure results in a map of $p$-values across the tested area, corrected for multiple comparisons based on the nonparametric null-distribution ([70] for further details). For each model comparison, we report the smallest $p$-value across the tested area, as an indicator of whether the given model significantly explains any neural data. In addition, for effect size comparison, we report $t_{max}$ for each comparison, the largest $t$-value in the significant ($p \leq 0.05$) area. For single-talker speech (Fig 2), this test included the whole cortex (as labeled by FreeSurfer). For subsequent tests of the 2-talker condition, the same tests were used, but the test area was restricted to the auditory ROI comprising the STG and transverse temporal gyrus in each hemisphere.

Initially, responses to speech in noise (Fig 3) were predicted from:

$$MEG \sim o_{mix} + o_{att} + o_{ign} + e_{mix} + e_{att} + e_{ign}$$

where $mix$ = mixture, $att$ = attended, and $ign$ = ignored. Masked onsets (Fig 4) were analyzed with the following:

$$MEG \sim o_{mix} + o_{att,overt} + o_{att,masked} + o_{ign,overt} + o_{ign,masked} + e_{mix} + e_{att} + e_{ign}$$

In order to test for a level-dependent nonlinear response to onsets in the mixture, this model was compared with the following:

$$MEG \sim o_{mix-low} + o_{mix-mid} + o_{mix-high} + o_{att,overt} + o_{att,masked} + e_{mix} + e_{att} + e_{ign}$$

where $mix$-$low$, -$mid$, and -$high$ = mixture low, mid, and high intensity. This model has the same number of predictors but assumes no awareness of onsets in the ignored speaker.

## STRF analysis

To evaluate STRFs, the corresponding model was refit with $T = [-100,...,500)$ milliseconds to include an estimate of baseline activity (because of occasional edge artifacts, STRFs are displayed between $-50$ to 450 milliseconds). Using the same 4-fold split of the data as for model fits, 4 STRF estimates were averaged, each using 1 fold of the data for validation and the remaining 3 for training. Because predictors and responses were $\ell1$ normalized for the reverse correlation, and STRFs were analyzed in this normalized space, STRFs provide an SNR-like measure of response strength at different latencies for each subject.

Auditory STRFs were computed for each subject and hemisphere as a weighted sum of STRFs in the auditory ROI encompassing the STG and transverse temporal (Heschl's) gyrus. Weights were computed separately for each subject and hemisphere. First, each source point was assigned a vector with direction orthogonal to the cortical surface and length equal to the total TRF power for responses to clean speech (sum of squares over time, frequency, and

predictor). The ROI direction was then determined as the first principal component of these vectors, with the sign adjusted to be positive on the inferior–superior axis. A weight was then assigned to each source as the dot product of this direction with the source's direction, and these weights were normalized within the ROI.

In order to make STRFs more comparable across subjects, they were smoothed on the frequency axis with a Hamming window of width 7 bins. STRFs were statistically analyzed in the time range [0,...,450) milliseconds using mass-univariate $t$-tests and ANOVAs, with $p$-values calculated from null distributions based on the maximum statistic ($t$, $F$) in 10,000 permutations [72].

For visualization and peak analysis, STRFs were upsampled to 500 Hz. Peak latencies were computed by first averaging auditory STRFs along the frequency axis and then finding the largest or smallest value in each subject's TRF in a window of [20, 200) milliseconds for single-speaker and stream-based analysis (Figs 2 and 3) or [20, 250) milliseconds for the masked onset analysis (Fig 4). Reported peak latencies are always average latencies across subject.

## Supporting information

**S1 Simulations. Simulations to assess TRF cross-contamination.** TRF, temporal response function.
(PDF)

**S1 Fig. MEG responses to clean speech, envelope only.** Spectrotemporal response function to the envelope spectrogram, when estimated without considering onsets. All other details are analogous to Fig 2E. Data in S4 Data.
(PDF)

**S1 Data. Data from Fig 2.** Model prediction accuracy maps and spectrotemporal response functions for plots shown in Fig 2. Data are stored as pickled Python/Eelbrain objects with corresponding meta-data.
(ZIP)

**S2 Data. Data from Fig 3.** Details as S1 Data.
(ZIP)

**S3 Data. Data from Fig 4.** Details as S1 Data.
(ZIP)

**S4 Data. Data from S1 Fig.** Details as S1 Data.
(ZIP)

**S5 Data. Data from S1 Simulation.** Details as S1 Data.
(ZIP)

## Acknowledgments

We would like to thank Shihab Shamma for several fruitful discussions and Natalia Lapinskaya for her help in collecting data and for excellent technical support.

## Author Contributions

**Conceptualization:** Christian Brodbeck, L. Elliot Hong, Jonathan Z. Simon.

**Data curation:** Christian Brodbeck.

**Formal analysis:** Christian Brodbeck, Alex Jiao.

**Funding acquisition:** L. Elliot Hong, Jonathan Z. Simon.

**Investigation:** L. Elliot Hong, Jonathan Z. Simon.

**Methodology:** Christian Brodbeck, Jonathan Z. Simon.

**Project administration:** Jonathan Z. Simon.

**Resources:** Jonathan Z. Simon.

**Software:** Christian Brodbeck.

**Supervision:** Jonathan Z. Simon.

**Validation:** Christian Brodbeck, Alex Jiao.

**Visualization:** Christian Brodbeck.

**Writing – original draft:** Christian Brodbeck.

**Writing – review & editing:** Christian Brodbeck, Alex Jiao, L. Elliot Hong, Jonathan Z. Simon.

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
