## [Editor Report · Decision Letter 0]

24 Jan 2020

Dear Dr Brodbeck, 

Thank you for submitting your manuscript entitled "Dynamic processing of background speech at the cocktail party: Evidence for early active cortical stream segregation" for consideration as a Research Article by PLOS Biology.

Your manuscript has now been evaluated by the PLOS Biology editorial staff, as well as by an academic editor with relevant expertise, and I'm writing to let you know that we would like to send your submission out for external peer review.

Please re-submit your manuscript within two working days, i.e. by Jan 28 2020 11:59PM.

Kind regards,

Roli Roberts

Senior Editor

PLOS Biology

---

## [Decision Letter · Decision Letter 1]

4 Mar 2020

Dear Dr Brodbeck,

Thank you very much for submitting your manuscript "Dynamic processing of background speech at the cocktail party: Evidence for early active cortical stream segregation" for consideration as a Research Article at PLOS Biology. Your manuscript has been evaluated by the PLOS Biology editors, an Academic Editor with relevant expertise, and by four independent reviewers.

You'll see that all four reviewers are broadly positive about the research question that you're addressing, but they also raise some significant concerns. For example, several of them question the focus of the paper and its relationship to the literature. Reviewers #2 and #3 think that you may have failed to exclude some alternative explanations for your observations (to the extent that rev #3 even contemplated rejection), while reviewers #1 and #4 want more robust assessment of model performance.

In light of the reviews (below), we will not be able to accept the current version of the manuscript, but we would welcome re-submission of a much-revised version that takes into account the reviewers' comments. We cannot make any decision about publication until we have seen the revised manuscript and your response to the reviewers' comments. Your revised manuscript is also likely to be sent for further evaluation by the reviewers.

We expect to receive your revised manuscript within 2 months. 

**IMPORTANT - SUBMITTING YOUR REVISION**

*Re-submission Checklist*

*Published Peer Review*

*PLOS Data Policy*

*Blot and Gel Data Policy*

Sincerely,

Roli Roberts

Senior Editor

PLOS Biology

REVIEWERS' COMMENTS:

Reviewer #1:

Brodbeck and colleagues present an experiment and its results on the neural processing of two-talker auditory scenes as measured with MEG. Participants were presented with two audio-books and asked to focus on one of them. In a first analysis, using the same analysis framework throughout the manuscript, the authors present support of using onset predictors in addition to envelope predictors. Next, modelling MEG signals with onset and envelope predictors at different stimulus frequencies for the mixed, attended and unattended speech, the authors show TRF models showing similarities and differences between coefficients of the onset and envelope pointing out differences in the shape and latency differences of high coefficients (peaks) between attended and ignored. In a final step, the stimuli are differently described; this time in terms of overt and masked onsets, separately for attended and ignored speech. Here, the authors find a longer latency for overt vs. masked onsets concerning a first peak in coefficients and a second peak of coefficients in attended speech that is absent for both overt and masked onsets for ignored speech. The authors suggest that this latency difference indicates active unmasking processes (similar to "filling in") in auditory cortex.

While the study is well designed and the analyses seem to be carefully performed, I have some concerns regarding the presentation of the study as well as the methodology. Main and minor points are listed below. 

Major points:

1. The analysis focuses mostly on the TRF models and does a good job in presenting the spread across participants in figures 2 to 4. However, such a strong (visual) focus on model coefficients underlines these outcomes while model performance seems "second stage". However, the focus of the effects should be on the model performance which provides a more convincing argument in this type of analyses as model coefficients are difficult to interpret (e.g., being inherently multivariate). Thus, it would be beneficial to provide figures on model performance as well (boxplot/violin plots + single participant indicators similar to 3D, 4C). Similarly, an analysis restricted to specific delay values (e.g., first, second, later peaks; similar to e.g. Puvvada and Simon, 2017) would make a better argument concerning their presence/absence. For latency changes, TRF models might be a good way pursue if one doesn't want to use shifting- window-like or single delay analyses.

2. The statistical testing and results reporting was unclear. This concerned both model performance comparisons as well as statistics on TRF coefficients. For example, I could not follow why, for comparing model performances from filtered MEG data optimized to sources in AC, TFCE is required. Why is a cluster analysis needed at this stage when the model performance is a single r-value that is compared between different models or with respect to a baseline? Also, what does t_max mean in this context? I see the use for the analysis of TRF coefficients but I don't see how analyses of model performance could benefit from it. In addition, the multiple comparison correction for the model coefficients remains unclear. I believe that the results are corrected as there is an indication in the final paragraph of the methods but that was not enough to fully follow the approach. 

3. While I like and agree with the conclusions drawn especially from the overt vs masked onset comparisons, I don't follow why this manuscript's title (and to a lesser degree the abstract and discussion) focuses to such a large extent on background speech processing rather than the masking results. In my opinion, the effects and conclusions on overt vs. masked onsets (amplitude and latency differences) constitute the main content being novel to the field while the absence of the second negative peak is interesting (it lends support to the hierarchical processing accounts discussed by the authors) seems rather like a strong but secondary result/conclusion. 

4. One of the main points of this paper is the comparison of onset and envelope descriptions of sound stimuli to explain MEG activity. While I agree with the conclusions and like how carefully these distinctions were analyzed and interpreted, I want to remind the authors that studies have already applied onset descriptions for the analysis of continuous sounds in similar analysis frameworks which is not mentioned in this study (e.g., Petersen et al., 2017, JNeurophysiol; Fiedler et al., 2017, JNeurEng, Fiedler et al., 2018, NeuroImage; Hausfeld et al., 2018, NeuroImage; and maybe more). Still, the explicit comparison and quantification of including onset features provided in this study provides a novel element. 

Minor points:

5. Related to point 2: figures 2D, 3B and 4B include grey-scale indicators of significance for specific delays, which are markers for significantly non-zero coefficients in any frequency band (l. 147/8). Were these already corrected for multiple comparisons (e.g., by removing small clusters not surviving a cluster-based thresholding) and was this performed across 2 dimensions (frequency and delay) or across the delays? While there is a lot of information conveyed in these graphs, simply indicating any significance across frequencies is a big simplification. How about adjusting the height of the significance bar according to the number of significant frequency bins? Another small point is that at the moment it is unclear which p-values are indicated: max, min, an average of some sort.

6. Ll. 185-188 and Fig 3C: This was difficult to follow.

7. It is suggested masked onsets of both attended and unattended speech undergo active processing to be "un-mask". Especially for unattended sounds this might be interesting for further investigation as it remains unclear with these data whether this unmasking works on single speech sources or more generally on the unattended sounds as a whole (representing attended vs all other sounds). Could you please comment on that in your response? 

8. The figures are difficult to read given their size, please make them bigger or maybe split them up such that single panels have more space. 

9. The color differences between minus and plus peaks are impossible to distinguish (maybe it was the print out but the colors seems to be very similar). Are the color differences between positive and negative peaks necessary? For the line graphs these peaks are easily discernible which also holds for the single participants markers in Fig. 3C

Reviewer #2:

This is an interesting paper that uses previously published MEG data from humans listening to connected speech, either as a single talker or two simultaneous talkers (one male, one female) with participants attending to one and ignoring the other. The study reports a difference in how "masked" onsets in speech are represented in the human auditory cortex, relative to onsets that are not masked by the competing talker. This difference is observed both for attended and unattended talkers. The authors suggest in the paper (including in the title) that this difference reflects an "active" process, and that the longer latency observed for the masked responses may reflect this active processing (or recovery) of the masked speech.

Although the premise is interesting, I do not believe that the authors have demonstrated what they claim. In particular the conclusions are based on linear systems analysis, which is applied to a highly nonlinear system. This approach of assuming linearity is almost universally adopted and has been a very useful first approximation, in part because speech is quite sparse, and so even mixtures of two talkers will having relatively limited spectrotemporal overlap. Nevertheless, the limitations of a linear-systems approximation to a nonlinear (and time-variant) system should not be overlooked. An important limitation is that a linear system assumes perfect superposition, whereas neural recordings do not demonstrate this. Indeed, the simplest auditory responses, from the Wave I, reflecting auditory-nerve responses, through brainstem responses (e.g., Wave V), to early cortical responses, all show changes in the morphology of the responses (amplitude and latency) based on the signal-to-noise ratio, ie, the degree to which a stimulus is masked. By this reasoning, it would be expected that partially masked sounds would produce smaller-amplitude and longer-latency responses, just as reported by the authors, not because of any active stream segregation, but because neural responses to partially masked sounds typically have smaller responses and longer latencies. Perhaps I am missing something important, which answers this criticism, and for this reason I have recommended revision rather than initial rejection. In any case, this seems an obvious alternative explanation, so it is not clear why the authors haven't addressed it, if only to rule it out.

Specific comments

The onset detection used by the authors seems to be based on a neural model. This needs a little more explanation, as well as more detail. For instance, what proportion of onsets overall were deemed "masked"? Also the measure itself (Max(0,(Max(attended,ignored) - mixture) could do with a little more explanation. Are the units in terms of the model's neural response? Does this measure have any influence on the size of the MEG responses reported in the figures?

Line 319. Early selective attention. The authors suggest that the masked responses show earlier manifestations of attention than the overt responses. But according to Fig. 4B, both the unmasked (overt) and the masked responses start to show significant effects of attention prior to 100 ms, and the difference seems to be more in the sign than in the magnitude of the difference. This needs to be clarified or reframed.

Line 439. Add ", respectively" after "Chapters 3 and 8" so that it's clear that the male read 3 and the female read 8.

Line 442. How was loudness matched, and what was the approximate sound pressure level of the two talkers? (Something more exact than "Comfortably loud")

Line 507. Typo: "am" should be "an".

Reviewer #3:

This manuscript presents research aimed at investigating the representation of attented and, especially, unattended speech in the auditory cortex. The authors ask subjects to attend to one of two concurrently stories at the same time while they record their MEG. Then they examine how well they can predict MEG, source localized to auditory cortex, from two different representations of the speech stimuli. One representation is a gammatone spectrogram of the stimulus (essentially eight envelopes corresponding to the energy in eight frequency bands). The second is a spectrogram of the acoustic onsets of the speech, again in eight bands, and derived using an auditory onset model. The goal is to test if early auditory cortex either conducts a fairly straightforward and stable spectrotemporal analysis of all input or carries out a more active process to that involves representing speech features from attended and unattended streams in a dissociable way. They reason that including the acoustic onset representations should allow them to answer this question because of the importance of onsets to auditory scene analysis and also because, analytically, the onsets in a mixture of two speech streams is not very predictable from the mixture of the onsets of the two streams. They report that… and conclude that…

Overall I thought this was a very interesting manuscript with a well conducted experiment and a sensible and sophisticated data analysis framework. I did have a few queries and comments for the authors though.

Main comments:

1) My first comment just speaks to the idea of making the rationale for the approach a bit more reader friendly in the introduction. I have to admit it took me two reads through the introduction for me to start to become comfortable with the ideas underlying the strategy. So, can I first clarify: the authors use the onset representation for two reasons: a) because of their importance for ASA, and b) because of this property that "the proportion of the variability in the mixture representations that cannot be predicted from the two sources is small for the envelopes, but substantially larger for the onsets". Is that accurate? If so, I wondered about giving readers a slightly more intuitive description of this. The way I have phrased it above is that "the onsets in a mixture of two speech streams is not very predictable from the mixture of the onsets of the two streams", at least less than for the envelope. Am I understanding what you are saying? Also, I was confused about how this links with the idea that "With acoustic onset features, interference is more pronounced because ongoing acoustic energy in one source can hide an increase in another source (see below for empirical verification of this claim)." Is this making the same point? It didn't seem so on first reading. And it was not at all obvious to me - I would have felt that envelopes were more likely to interfere with each other because they are more continuous. You say you empirically verify this below - is that in Figure 3C). Anyway, the reason I am commenting on this, is that when I got to the results I just had this nagging feeling that the delay I was seeing in the STRFs for the individual sources might just be a signal processing glitch. Does the supplementary figure answer that specific concern? Or is that only to provide evidence that the ignored speech source contribution cannot be explained from the mixture. It feels like only the latter to me. 

2) Related to the previous point, is it also fair to say that the current analysis basically seeks to overturn the claims made in the Puvvada & Simon (2017) paper - based on an analysis that is more sensitive to the neurophysiological responses to the individual speech streams?

3) Given recent (cited) work on the different effects of attention in HG vs STG (O'Sullivan et al., 2019), I also wondered about the auditory localization used in the present study. I think I understand that the present study does not contradict the O'Sullivan paper. It merely suggests that both attended and unattended speech are separably represented in early auditory cortex, not that both are modulated strongly by attention in early auditory cortex. Nonetheless, I wondered about how confident we can be that we are really looking early auditory cortex not contamination by strong attention effects from, for example, STG. I guess it all comes down to the latency of the effects you are seeing? The source onset STRFs in Figure 3B show strong attentional modulations of a later component around 150 ms - might this be STG? And so we are safe to assume everything before 100 is from a "lower" cortical area than STG?

4) I also wondered about how much the difference in performance between onset STRFs and envelope STRFs might be due to the choice to only have 8 frequency bands in the representation. I wonder might the authors care to wax lyrical on whether or not this result is likely to still be true if one used, say, 100 bands. Might that be enough spectral resolution for us to see these effects also in the envelope STRFs (assuming it was possible to fit the models reliably)? 

Minor comments:

1) I wondered why the STRF to the mixture envelope was not plotted in Figure 3 - just for the sake of completeness.

2) For people who have not read Bob Carlyon's 2004 paper, I thought it might be nice to slightly unpack the following sentence "Alternatively, the auditory cortex could employ an active process to dynamically recover and represent potential speech features regardless of what stream they belong to, and in doing so provide selective attention with more elaborate representations [11]."

3) The value of the un-masking section on page 16 - and indeed the whole paper really - was brought home to me when I read the line about auditory cortex potentially filling in the underlying speech sources. I thought that was a nice way to think about what kind of thing cortex might be doing in terms of active segregation. I wonder might the authors want to move that point earlier and/or unpack it a small bit for folks who are not familiar (as I happened to be) with references 26 and 27. Just a suggestion.

Reviewer #4:

This study examines how the foreground and background speech streams are processed when presented simultaneously. Using MEG recordings and linear STRF modeling, it examines how the responses to the onset and envelope of the sources are represented. The results suggest that when the spectro-temporal features mask each other, an active process extracts the onsets of the sources, resulting in a MEG activation that is delayed by 20 ms compared to the activation in response to overt (unmasked) sources. These findings are novel and significant, identifying the brain mechanisms that underlie speech perception in complex situations. It is technically sound and the experiment is well designed. The statistical analysis appropriate and supplementary information is useful. Also, the data needed to replicate the study are available.

There is only one major issue I think needs to be addressed. The authors use linear modeling based on information extracted from onsets and envelops of the signals. A primary conclusion is that onsets are critical for many aspects of the processing. However, the model does not allow too many alternatives to that, as the predictors are only based on onsets or on envelopes. I think the conclusions (including those in the supplement) would be much stronger if offset detectors were built into the model, equivalent to the onset detectors, and if it was shown that no model improvement is obtained from predictors based on the offsets, or, that the offsets themselves cannot predict the results. So, I think it is critical to do additional modeling with the offsets considered before the conclusions of the modeling can be presented as they stand.

Minor issues:

L36: in "… an acoustic scene, the acoustic signal …" the word "acoustic" is repetitive. Saying "… an acoustic scene, the signal …" is sufficient.

L507 am -> an

---

## [Decision Letter · Decision Letter 2]

13 Aug 2020

Dear Dr Brodbeck,

Thank you for submitting your revised Research Article entitled "Neural speech restoration at the cocktail party: Auditory cortex recovers masked speech of both attended and ignored speakers" for publication in PLOS Biology. I have now obtained advice from the original reviewers and have discussed their comments with the Academic Editor.

IMPORTANT: You'll see that reviewer #2 continues to express deep scepticism about your findings, and there is also a cautious note from reviewer #3. However, after discussing the reviewers' comments with the Academic Editor, we have decided to consider your manuscript further if you make these limitations clear to the readers. Specifically, the Academic Editor would like you to further consider and discuss the issue of the SNR as mentioned by reviewer #2. You should also address the remaining requests from revs #1 and #3.

Based on the reviews, we will probably accept this manuscript for publication, assuming that you will modify the manuscript to address the remaining points raised by the reviewers. Please also make sure to address the Data and other policy-related requests noted at the end of this email.

We expect to receive your revised manuscript within two weeks. Your revisions should address the specific points made by each reviewer. In addition to the remaining revisions and before we will be able to formally accept your manuscript and consider it "in press", we also need to ensure that your article conforms to our guidelines. A member of our team will be in touch shortly with a set of requests. As we can't proceed until these requirements are met, your swift response will help prevent delays to publication.

*Copyediting*

*Published Peer Review History*

*Early Version*

*Submitting Your Revision*

Sincerely,

Roli Roberts

Senior Editor,

rroberts@plos.org,

PLOS Biology

ETHICS STATEMENT:

-- Please include the full name of the IACUC/ethics committee that reviewed and approved the animal care and use protocol/permit/project license. Please also include an approval number.

-- Please include the specific national or international regulations/guidelines to which your animal care and use protocol adhered. Please note that institutional or accreditation organization guidelines (such as AAALAC) do not meet this requirement.

-- Please include information about the form of consent (written/oral) given for research involving human participants. All research involving human participants must have been approved by the authors' Institutional Review Board (IRB) or an equivalent committee, and all clinical investigation must have been conducted according to the principles expressed in the Declaration of Helsinki.

DATA POLICY:

We note that your raw data are made available in an institutional repository. While this is accessible and clearly laid out, ideally (for long-term robustness) we would prefer it to be hosted on a non-institutional repository (e.g. Dryad, Figshare, Github), and ask you to explore that.

In addition, we ask that all individual quantitative observations that underlie the data summarized in the figures and results of your paper be made available in one of the following forms:

Regardless of the method selected, please ensure that you provide the individual numerical values that underlie the summary data displayed in the following figure panels as they are essential for readers to assess your analysis and to reproduce it: Figs 2BCDE, 3ABCD, 4ABCD and S1, S2. NOTE: the numerical data provided should include all replicates AND the way in which the plotted mean and errors were derived (it should not present only the mean/average values).

REVIEWERS' COMMENTS:

Reviewer #1:

This work by Brodbeck, Jiao, Hong and Simon is a resubmission of a previously reviewed manuscript. The current version of the manuscript addressed most of the concerns of the initial version such that it reads stronger and provides clearer links between results and claims. I very much appreciate the reanalysis and introduction of model fit statistics here with its non-trivial creation of permutated data/predictors as well as the extensive work on the readability of text and figures. However, some comments and unclear points (mainly on the statistics) remained and are listed below.

1. Model fit statistics I. Although consistent and significant, the z-values and corresponding r-values for improvement seem to be small and around delta_z = .01 in Fig 2D and otherwise lower than ˜.005 in Figs.3B and 4B. What do you think is the reason for this? Could it be due to correlations between predictors and/or small effect sizes in general? One suggestion might be to indicate the z- or r-value obtained with the tested model for better judgement.

2. Model fit statistics II. I just want to be sure to exactly follow the approach in combination with the source localization. I assume that the dots in Figs 2D, 3B, 4B depict the individual improvements when including a certain predictor vs. not including it and show the average improvement across the x dipole models in each of the hemisphere. When considering the single speaker condition, are these values showing the difference of, for example, the models with onset and envelope vs. models with only the onset to infer the significance of the envelope in this case (this is my guess)? Or do you test the model with envelope vs a null model that is established with a noise simulation (ref 72)?

3. l. 326-331. It is investigated whether masked onsets are delayed. However, only descriptive statistics are provided. For better support of the claim, please test for this formally. This comes back in l.382 where it is stated as a summary of masking results that "[these] latency shifts increase at successive stages", which requires a comparison of the latency shifts between earlier and later window.

4. Unlike for the t_max it remains unclear to me what the p-values that are mentioned (e.g., ll. 142, 208, …) denote, which is mainly due to the source modeling of signals in each vertex of the modeled brain surface (also in comments on the first version). Are these the corresponding p-values of the statistic of the spatial cluster showing highest significance (as there might be several significant clusters within one hemisphere) or something else? The explanation that "group-level statistics were evaluated with spatial permutation tests" (l. 204) didn't help with better understanding and might benefit from a (very short) description of the procedure outlined in ref 72.

5. Just to be sure this comment is not misunderstood: the following is a final, constructive and, by all means, friendly remark on the topic of masked speech. While re-reading the manuscript and focusing on the masking aspect, I was reminded of Fiedler et al. (2019, NeuroImage) (ref 21). Their results presented in their figure 3 (contrasting relative SNR changes in attended vs ignored speakers) look very similar to the results presented in this manuscripts Figs 4D and E and indicate a similar latency shift (which was not tested for or discussed in Fiedler et al., 2019). It's great and encouraging to see convergence here for different types of data, stimuli and languages (disclaimer: neither am I one of the authors nor am I in any way affiliated with this lab or have any other interests).

Reviewer #2:

My main concern when reading the previous version of the manuscript was that the authors had not supported their primary claim that the cortex "recovers masked speech". Instead, I suggested that the results could be explained by the fact that responses to partially masked onsets (i.e., onsets presented at a low signal-to-noise ratio) would be expected to be lower in amplitude and have a longer latency, based on what we already know about auditory responses to stimuli in noise, from the earliest stages of auditory processing. Indeed, the only justification for calling it "recovery" is that the authors' neural model of onset detection does not detect partially masked onsets in the same way as the measured responses from human cortex seem to do. But this could easily be a problem with their very simple model, rather than any insight into how the auditory cortex processes sound.

The authors now acknowledge this problem, and counter it by providing additional analyses to show that their model (incorporating the overt vs. masked distinction) outperforms a model that incorporates information about the instantaneous target level (quantized into 3 level regions) but ignores the signal-to-noise ratio (SNR). However, this approach does not address the problem in a satisfactory way: The effect of nonlinearity goes beyond just producing level-dependent effects: Nonlinearity produces interactions when two or more stimuli are combined. We already know that at low SNRs (as is the case here, where the average SNR is 0 dB), SNR will modulate the responses to a target sound more strongly than simple intensity. It has nothing to do with "cortical recovery" of signals. Thus, the additional analysis does not rule out, or even reduce, the possibility that what the authors have measured is simply the expected outcome of stimuli that overlap acoustically, producing partial masking.

In summary, the major potential problem raised in the first round of the reviews seems confirmed: the results are predictable based simply on the fact that neural responses to onsets will be reduced (and slightly delayed) when the onsets are partially masked. This type of response is well established at many stages of the auditory pathways and does not provide evidence for "recovery" or any other type of active cortical processing. Indeed the time course in itself (with latencies of less than 100 ms) should make the authors (and readers) suspicious of the claim that any active recovery process is undertaken.

With this interpretation, most of the speculation provided in the Discussion (Lines 416-494) and the unhelpful black-box model in Fig. 5 become irrelevant. As the data themselves are not new and were already published elsewhere, there seems little else to justify publishing the current study.

Reviewer #3:

Many thanks to the authors for their extensive efforts in addressing my previous comments. I think the new introduction is much clearer. Most of my other comments have been fully addressed also.

I guess I still have a little bit of residual unease about the analysis of masked vs overt onsets. You have determined which onsets are masked and which are not based on your own particular choices of how to represent the acoustic onsets (i.e., using your auditory edge detection model with particular parameters). However, it is not obvious that these onsets will be genuinely masked from the perspective of the cochlear output. If they are not, then should we really be considering them as masked and overt? And if they are, then I still struggle to understand how one can confidently model their separate contributions to the neural responses. I do appreciate that the simulations in Fig. S1 and S2 are aimed to ease our worries on this front – and they achieve that goal mostly. But, I guess, I still feel like we are in a bit of a Catch-22. If they are not perfectly masked, then what does it even mean to call them masked? And, if they are perfectly masked, then how is it possible to separate them at all? If you could help clarify this for me, I would greatly appreciate it.

Also, and I hesitate to suggest this given the amount of work you have already done, but I wondered about the simulations in S1 and S2. Might it not have been more compelling to allow non-zero ignored TRFs and show that there was no warping/delaying of those TRF peaks by the coincidence of the attended and ignored (masked) onsets. Maybe not – the simulation you already have seems fairly compelling. So maybe a general answer to my first comment would suffice.

Reviewer #4:

The revised version of the manuscript sufficiently addressed all of my concerns. I do not have any more suggestions.

---

## [Editor Report · Decision Letter 3]

14 Sep 2020

Dear Dr Brodbeck,

On behalf of my colleagues and the Academic Editor, Manuel S. Malmierca, I am pleased to inform you that we will be delighted to publish your Research Article in PLOS Biology. 

Early Version

PRESS 

Kind regards,

Vita Usova

Publication Assistant, 

PLOS Biology

on behalf of

Roland Roberts,

Senior Editor

PLOS Biology